# TRAIP drives replisome disassembly and mitotic DNA repair synthesis at sites of incomplete DNA replication

**Remi Sonneville[1†], Rahul Bhowmick[2†], Saskia Hoffmann[3], Niels Mailand[3]\*, Ian D Hickson[2]\*, Karim Labib[1]\***

[1]MRC Protein Phosphorylation and Ubiquitylation Unit, School of Life Sciences, University of Dundee, Dundee, United Kingdom; [2]Department of Cellular and Molecular Medicine, Center for Chromosome Stability, University of Copenhagen, Copenhagen, Denmark; [3]Novo Nordisk Foundation Center for Protein Research, Faculty of Health and Medical Sciences, University of Copenhagen, Copenhagen, Denmark

**Abstract** The faithful segregation of eukaryotic chromosomes in mitosis requires that the genome be duplicated completely prior to anaphase. However, cells with large genomes sometimes fail to complete replication during interphase and instead enter mitosis with regions of incompletely replicated DNA. These regions are processed in early mitosis via a process known as mitotic DNA repair synthesis (MiDAS), but little is known about how cells switch from conventional DNA replication to MiDAS. Using the early embryo of the nematode *Caenorhabditis elegans* as a model system, we show that the TRAIP ubiquitin ligase drives replisome disassembly in response to incomplete DNA replication, thereby providing access to replication forks for other factors. Moreover, TRAIP is essential for MiDAS in human cells, and is important in both systems to prevent mitotic segregation errors. Our data indicate that TRAIP is a master regulator of the processing of incomplete DNA replication during mitosis in metazoa.

DOI: https://doi.org/10.7554/eLife.48686.001

**\*For correspondence:**
niels.mailand@cpr.ku.dk (NM);
iandh@sund.ku.dk (IDH);
kpmlabib@dundee.ac.uk (KL)

[†]These authors contributed equally to this work

**Competing interests:** The authors declare that no competing interests exist.

## Introduction

Chromosome duplication in eukaryotes is initiated from many origins of DNA replication on each chromosome, leading to the progression of DNA replication forks across the genome until fork convergence induces DNA replication termination. However, in species with genomes larger than about 100 Mb, it becomes increasingly likely that a small number of very late-replicating loci will not have completed DNA replication when cells enter mitosis (*Al Mamun et al., 2016*).

Metazoa have evolved pathways that quickly process sites of incomplete replication during mitosis, in order to facilitate the completion of chromosome segregation before cell division (*Bhowmick and Hickson, 2017*; *Moreno et al., 2016*). In human cells, factors such as the FANCD2 protein mark sites of unreplicated DNA, and then SLX4-dependent nucleolytic cleavage in mitosis promotes RAD52-dependent DNA repair synthesis (*Bhowmick et al., 2016*; *Minocherhomji et al., 2015*). Loci prone to such breakage and repair, particularly in response to DNA replication stress, include common fragile sites that are defined cytologically as breaks or gaps in metaphase chromosomes (*Glover et al., 2017*). Nevertheless, the mechanism by which mitotic processing initiates at such sites is not well understood. In particular, little is known about how conventional DNA replication forks are converted during mitosis into sites of DNA repair synthesis. One possibility is that the processing of sites of incomplete DNA replication begins with disassembly of the replisome, which otherwise protects the parental DNA at replication forks from attack by nucleases. The replisome is

assembled at origins of replication once per cell cycle, and is normally disassembled during DNA replication termination (*Dewar and Walter, 2017*; *Maric et al., 2014*; *Moreno et al., 2014*). Replisome disassembly is an active process, which is initiated by ubiquitylation of the CMG helicase via the ubiquitin ligase CUL-2$^{LRR-1}$ (*Dewar et al., 2017*; *Sonneville et al., 2017*), leading to disassembly by the Cdc48 segregase and its essential co-factors Ufd1 and Npl4 (*Franz et al., 2011*; *Maric et al., 2017*).

Using the early embryo of *C. elegans* as a model system, we previously identified a mitotic pathway for CMG disassembly that also requires CDC-48_UFD-1_NPL-4, and is able to remove any post-termination replisomes that escape the normal disassembly process during S-phase (*Sonneville et al., 2017*). This mitotic pathway for replisome disassembly requires a ubiquitin ligase known as TRAIP (*Deng et al., 2019*; *Priego Moreno et al., 2019*), which is important for genome integrity in human cells (*Feng et al., 2016*; *Harley et al., 2016*; *Hoffmann et al., 2016*; *Soo Lee et al., 2016*) and is required to remove the replisome from converged DNA replication forks during the repair of inter-strand DNA crosslinks in extracts of *Xenopus laevis* eggs (*Wu et al., 2019*).

Very recently, it was shown that addition of mitotic cyclin-dependent kinase to *Xenopus* egg extracts induces TRAIP-dependent replisome disassembly (*Deng et al., 2019*; *Priego Moreno et al., 2019*), even when DNA replication is incomplete, and leads to fork breakage and DNA rearrangements (*Deng et al., 2019*). However, it has yet to be determined whether TRAIP is important in metazoan cells and tissues for replisome disassembly in response to incomplete DNA replication, and whether TRAIP is required for mitotic DNA repair synthesis at common fragile sites. Here, we address these issues, using an integrated approach that combines the advantages of the *C. elegans* early embryo and human cell lines as model systems.

## Results and discussion

### Replisome disassembly after incomplete DNA replication in the *C. elegans* early embryo requires the CDC-48_UFD-1_NPL-4 segregase

Monitoring replisome disassembly during mitosis is difficult in eukaryotic cells that have been exposed to DNA replication stress, since the S-phase checkpoint response usually blocks mitotic entry under such conditions. However, the *C. elegans* early embryo provides a useful model system in this regard, since the checkpoint response to replication stress is attenuated, such that mitotic entry is delayed but not blocked (*Brauchle et al., 2003*; *Encalada et al., 2000*). Moreover, the transparent nature of the embryo facilitates live cell imaging.

We previously established a live cell assay for replisome disassembly in the first cell cycles of the *C. elegans* embryo (*Sonneville et al., 2017*), based on monitoring the presence or absence of a GFP-tagged subunit of the CMG helicase on condensing chromatin (*Figure 1A*; chromatin is visualised via mCherry-tagged histone H2B). To assay for the presence of the replisome on mitotic chromatin after subjecting cells to DNA replication stress, we depleted the RPA-1 component of the RPA single-strand binding protein complex by RNAi (*Figure 1B*, *Figure 1—figure supplement 1A*). As shown previously (*Sonneville et al., 2015*), chromosome condensation during prophase of the first mitotic cell cycle was impaired after depletion of RPA-1 (*Figure 1C*, *Figure 1—figure supplement 1B*), reflecting mitotic entry in the presence of incomplete DNA replication. The chromatin subsequently aligned on the metaphase plate, but anaphase was impaired due to multiple chromatin bridges (*Figure 1C*, *Figure 1—figure supplement 1B*; compare *rpa-1* RNAi with control). Strikingly, components of the CMG helicase, such as the GINS subunits PSF-1 and SLD-5, and the CDC-45 protein, were not detected on the partially replicated chromatin during mitosis (*Figure 1C–D*, metaphase; *Figure 1—figure supplement 1C–D,F–H*). This reflected active disassembly of the CMG helicase by the CDC-48_UFD-1_NPL-4 segregase, since CMG components were retained on mitotic chromatin when NPL-4 and RPA-1 were depleted simultaneously (*Figure 1B–D*; *Figure 1—figure supplement 1A,C–D,F–H*; note that *npl-4* depletion also induced chromatin bridges, reflecting the multiple roles of CDC-48_UFD-1_NPL-4 during the cell cycle, including during mitosis). These data indicate that the replisome is disassembled before metaphase when DNA replication is perturbed in the *C. elegans* early embryo.

To confirm the link between replisome disassembly and incomplete DNA replication, we depleted ribonucleotide reductase via RNAi against *rnr-1* and thus depleted dNTPs. This produced a

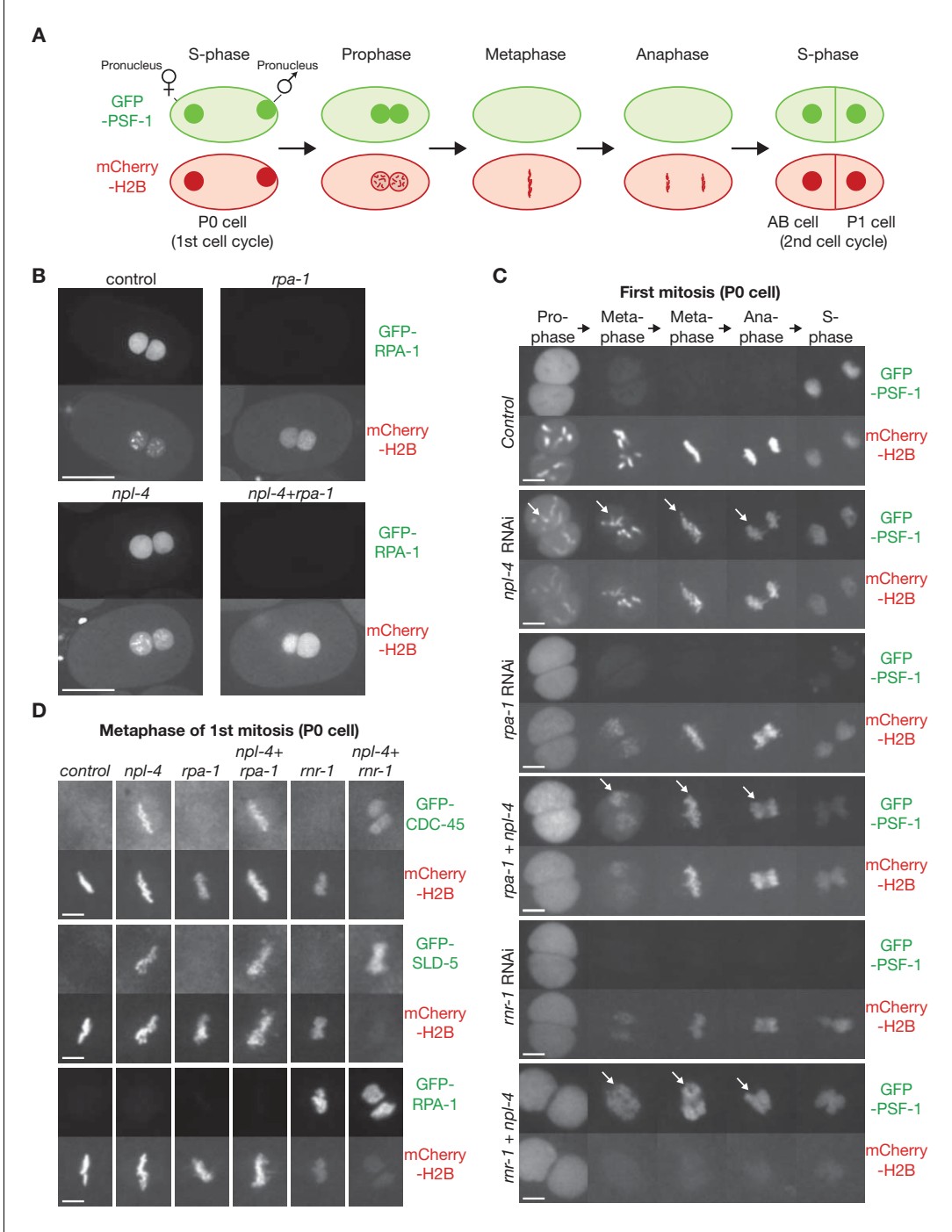

**Figure 1.** Incomplete DNA replication in *C. elegans* early embryos leads to unloading of the CMG helicase from chromatin by the CDC-48_UFD-1_NPL-4 segregase. (**A**) Illustration of the first cell cycle in the *C. elegans* early embryo. (**B**) Worms expressing GFP-RPA-1 and mCherry-Histone H2B were exposed to the indicated RNAi treatments (control = no RNAi). Images of whole embryos are shown, during prophase of the first embryonic cell cycle (two minutes before nuclear envelope breakdown), in order to illustrate the efficiency of RPA-1 depletion. (**C**) Worms expressing GFP-PSF-1 and mCherry-Histone H2B were exposed to the indicated RNAi treatments, before time-lapse imaging of the first mitosis in isolated embryos. The arrows indicate persistence of CMG on mitotic chromatin. (**D**) Worms expressing the indicated GFP-tagged replication proteins, together with mCherry-Histone H2B, were exposed to the same range of RNAi treatments as in (**C**). Examples are shown of the metaphase stage of the first embryonic mitosis. Scale bars = 20 µm in (**B**) and 5 µm in (**C and D**). Quantification of microscopy data is provided in *Figure 1—figure supplement 1*.

DOI: https://doi.org/10.7554/eLife.48686.002

The following figure supplements are available for figure 1:

*Figure 1 continued on next page*

*Figure 1 continued*

**Figure supplement 1.** Quantification of *C. elegans* microscopy data in *Figure 1*.
DOI: https://doi.org/10.7554/eLife.48686.003

**Figure supplement 2.** Condensation is profoundly defective following depletion of RNR-1 and NPL-4 in *C. elegans* early embryos.
DOI: https://doi.org/10.7554/eLife.48686.004

profound defect in DNA replication (*Sonneville et al., 2015*), as reflected by greatly impaired chromosome condensation during prophase (*Figure 1C*, *Figure 1—figure supplement 1B*; *Figure 1—figure supplement 2A–B* show that condensin II was not detected on mitotic chromatin after RNR-1 depletion). RPA accumulated on metaphase chromatin under such conditions (*Figure 1D*; *Figure 1—figure supplement 1F*), and chromosome segregation during anaphase was severely defective, thus blocking the subsequent reformation of the nuclear envelope (*Figure 1C*; 5 of 5 embryos examined behaved the same). CMG components were not detected on mitotic chromatin following *rnr-1* RNAi, mirroring the defect seen after depletion of RPA-1, and this was due to active disassembly of the replisome by CDC-48_UFD-1_NPL-4, since CMG components persisted on mitotic chromatin in embryos treated with *rnr-1 npl-4* double RNAi (*Figure 1C–D*, *Figure 1—figure supplement 1C–D, F–H*).

Interestingly, the mCherry-H2B signal was barely detectable throughout mitosis upon co-depletion of RNR-1 and NPL-4 (*Figure 1C–D*, *Figure 1—figure supplement 1E*), and this reflected the absence from mitotic chromatin of both condensin II and condensin I (*Figure 1—figure supplement 2*; note that condensin II normally associates with mitotic chromatin during prophase, whereas condensin I is loaded during metaphase). Most likely, the persistence of the CMG helicase at replication forks supported continued slow progression of the replisome throughout the genome, uncoupled from DNA synthesis, thereby unwinding the parental DNA duplex and displacing parental histones from unreplicated chromatin. In contrast, continued progression of the replisome in the presence of replication stress might be more limited following double RNAi to *rpa-1* and *npl-4*, such that a significant amount of unreplicated chromatin (marked by mCherry-H2B) persists across the genome. This issue is explored further below.

## TRAIP is dispensable for proliferation in *C. elegans*, but preserves genome integrity in combination with CUL-2[LRR-1]

To investigate whether the TRAIP ubiquitin ligase is required for mitotic replisome disassembly and the preservation of genome integrity in response to incomplete DNA replication in the *C. elegans* early embryo, we used CRISPR-Cas9 to delete the *trul-1* gene (*Figure 2—figure supplement 1*; TRUL-1 = TRAIP Ubiquitin Ligase 1). Interestingly, homozygous *trul-1Δ* worms are viable, in contrast to the lethality of previously reported deletions of TRAIP orthologues in the mouse and in *Drosophila* (*Merkle et al., 2009*; *Park et al., 2007*), probably reflecting the small number of cell divisions in the *C. elegans* life cycle and the relatively small size of the worm genome (around 100 Mb, compared to about 180 Mb in *Drosophila* and 1,500 Mb in mouse). Cell cycle progression appeared normal in homozygous *trul-1Δ* embryos (see below, *Figure 2—figure supplement 4*), but RNAi depletion of LRR-1 in *trul-1Δ* worms caused the persistence of the CMG helicase on mitotic chromatin (*Figure 2—figure supplement 2A–B*). A similar observation was made using *lrr-1 trul-1* double RNAi (*Deng et al., 2019*), confirming that TRUL-1 is required for the mitotic pathway that disassembles post-termination replisomes. Strikingly, LRR-1 depletion in *trul-1Δ* worms led to the appearance of anaphase DNA bridges in the early embryo (*Figure 2—figure supplement 2C*), indicating that the S-phase and mitotic pathways for replisome disassembly are jointly important for the preservation of genome integrity.

## TRAIP is required for replisome disassembly in response to DNA replication stress

By exposing control and *trul-1Δ* worms to *rpa-1* RNAi, we observed that CMG persisted on mitotic chromatin when RPA-1 was depleted in the absence of TRAIP (*Figure 2A*, *Figure 2—figure supplement 3A*). In contrast, *rpa-1 lrr-1* double RNAi did not prevent replisome disassembly in wild type worms, and was equivalent to *rpa-1* single RNAi (*Figure 2A*, *Figure 2—figure supplement 3A*).

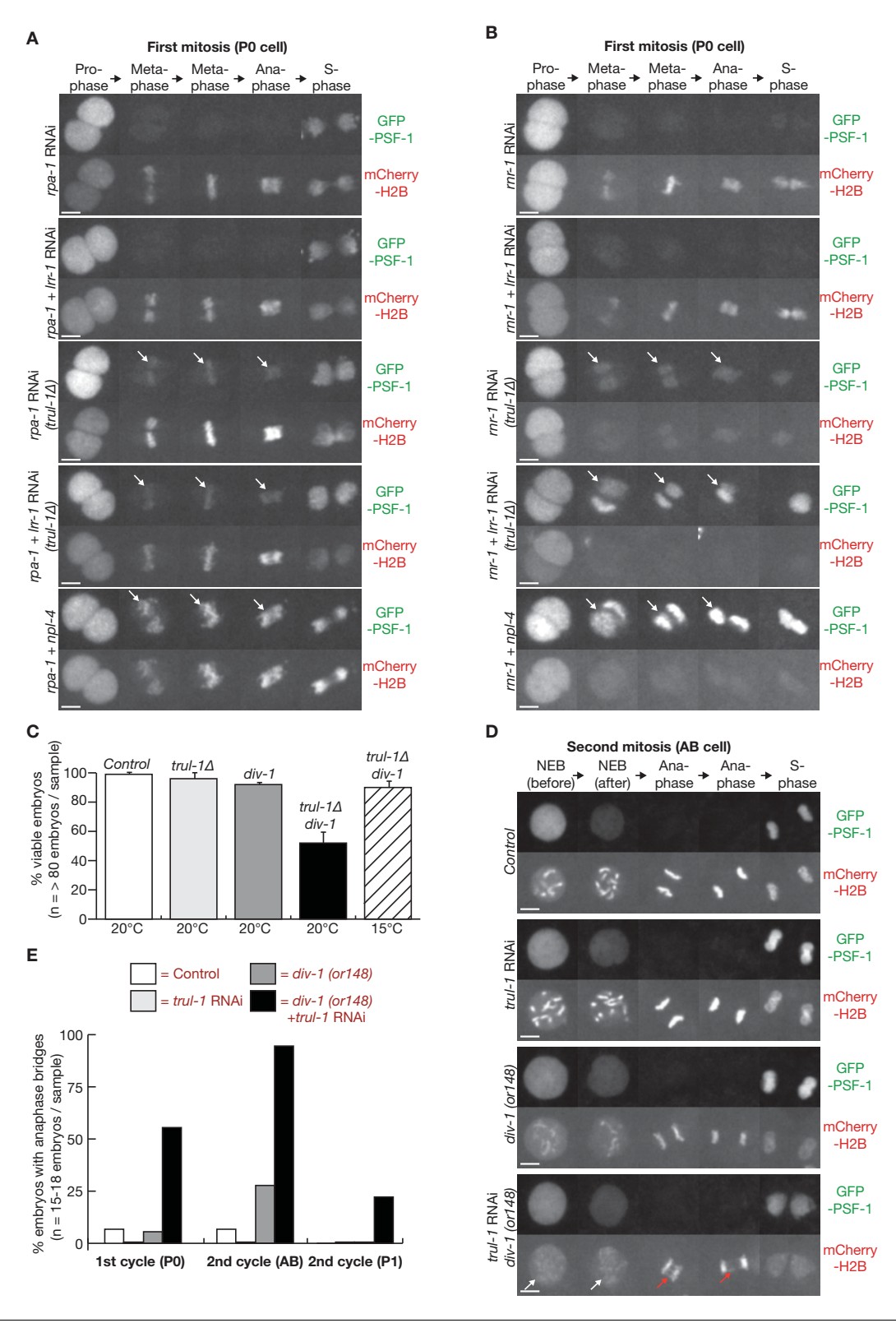

**Figure 2.** The TRUL-1 orthologue of human TRAIP is required for CMG unloading and genome integrity in response to incomplete DNA replication in *C. elegans* early embryos. (**A**) Videos of control and *trul-1Δ* embryos expressing GFP-PSF-1 and mCherry-Histone H2B, treated with RNAi to *rpa-1*, *lrr-1* and *npl-4* as indicated. Arrows indicate persistence of CMG on mitotic chromatin. (**B**) Videos of control and *trul-1Δ* embryos expressing GFP-PSF-1 and mCherry-Histone H2B, treated with the indicated RNAi treatments. Arrows indicate persistence of CMG on mitotic chromatin. (**C**) Embryonic viability

*Figure 2 continued on next page*

*Figure 2 continued*

was measured for the indicated strains, using worms grown at the temperatures shown. The data correspond to the mean and standard deviation from three independent experiments. (D) Videos of control and *div-1(or148)* temperature sensitive embryos expressing GFP-PSF-1 and mCherry-Histone H2B, treated with empty RNAi vector or RNAi to *trul-1* as indicated. In order to inactivate DIV-1, the worms were grown at the restrictive temperature of 25°C for 24 hr before imaging. The images correspond to division of the AB cell during the second embryonic cell cycle. White arrows indicate defective chromosome condensation during prophase, whereas red arrows denote chromatin bridges during anaphase. (E) Quantification of the percentage of cells with anaphase bridges in the first two embryonic cell cycles, for the experiment shown in (D). Scale bars in all microscopy images = 5 µm. Further quantification of microscopy data is provided in *Figure 2—figure supplement 3*.

DOI: https://doi.org/10.7554/eLife.48686.005

The following figure supplements are available for figure 2:

**Figure supplement 1.** Deletion of the *C. elegans trul-1* gene by CRISPR-Cas9.

DOI: https://doi.org/10.7554/eLife.48686.006

**Figure supplement 2.** TRUL-1 is required for disassembly of the CMG helicase during mitosis, and preserves genome integrity together with LRR-1.

DOI: https://doi.org/10.7554/eLife.48686.007

**Figure supplement 3.** Quantification of *C. elegans* microscopy data in *Figure 2*.

DOI: https://doi.org/10.7554/eLife.48686.008

**Figure supplement 4.** Timing of cell division for the first two embryonic cell cycles after inactivation of DIV-1 and TRUL-1.

DOI: https://doi.org/10.7554/eLife.48686.009

**Figure supplement 5.** TRUL-1 is important for genome integrity and the successful completion of chromosome replication, when cells are exposed mild DNA replication stress, resulting from a diluted *rnr-1* RNAi.

DOI: https://doi.org/10.7554/eLife.48686.010

These data indicated that replisome disassembly following RPA-1 depletion requires TRAIP, but is independent of the CUL-2$^{LRR-1}$ pathway that acts when two converging replication forks encounter each other during DNA replication termination. Similarly, we observed that the CMG helicase accumulated on mitotic chromatin following *rnr-1* RNAi in *trul-1Δ* worms. However, this effect was enhanced in combination with *lrr-1* RNAi (*Figure 2B*, *Figure 2—figure supplement 3B*; compare *rnr-1* RNAi and *rnr-1 lrr-1* double RNAi in *trul-1Δ*). Moreover, mCherry-H2B on condensing chromatin was barely detectable throughout mitosis after *rnr-1 lrr-1* double RNAi in *trul-1Δ* worms, equivalent to the effect of *rnr-1 npl-4* double RNAi as discussed above (*Figure 2B*, *Figure 2—figure supplement 3B*). These findings are consistent with the notion that TRUL-1 normally mediates replisome disassembly following *rnr-1* RNAi, probably during mitosis (equivalent to TRAIP-dependent mitotic disassembly of post-termination replisomes). However, continued replisome progression in the absence of TRUL-1 allows some forks to converge, thereby leading to disassembly via the CUL-2$^{LRR-1}$ pathway.

## TRAIP preserves genome integrity and promotes survival in response to DNA replication stress

To investigate the physiological significance of the TRUL-1 pathway in response to incomplete DNA replication, we needed to use a milder form of DNA replication stress, since *rpa-1* and *rnr-1* RNAi both cause profound replication defects that prevent the successful completion of the first embryonic cell cycle. Initially, we used worms with a point mutation in the *div-1* gene, encoding the second-largest subunit of DNA polymerase alpha, since previous work showed that *div-1(or148ts)* worms are inviable at 25°C but can grow at lower temperatures (*Encalada et al., 2000*). Whereas the viability of *div-1* or *trul-1Δ* single mutants was close to that of the wild type control at the semi-permissive temperature of 20°C, the combination of *trul-1Δ* with *div-1* at 20°C produced synthetic lethality in around 50% of embryos. This synthetic lethality was suppressed by growth at the permissive temperature of 15°C (*Figure 2C*). These findings indicated that *C. elegans* TRAIP is important for worms to survive DNA replication stress, though TRUL-1 is dispensable in unperturbed cell cycles. It was not possible to compare *lrr-1* RNA with *trul-1Δ* in similar experiments, since depletion of LRR-1 is itself sufficient to produce DNA replication stress and lethality (*Merlet et al., 2010*).

To examine in greater detail how TRAIP contributes to the viability of *div-1* worms experiencing replication stress, we monitored the formation of anaphase chromatin bridges in the first two embryonic cell cycles of control or *div-1(or148ts)* worms grown at the restrictive temperature of 25°C, after treatment with *trul-1* RNAi or empty vector. As shown in *Figure 2D–E*, depletion of TRUL-1 led to

an accumulation of mitotic chromatin bridges in the *div-1* mutant (red arrows in *Figure 2D*), indicating that TRAIP is important for chromosome integrity in response to replication stress. In the second embryonic cell cycle, the effect was greatest (around 90% of embryos, n = 18) in the anterior AB cell (*Figure 1A*), which is known to have a weak S-phase checkpoint and thus is vulnerable to entering mitosis with persistent DNA replication stress (*Brauchle et al., 2003*). Conversely, chromatin bridges were observed less frequently in the posterior P1 cell. This cell has an enhanced checkpoint response (*Brauchle et al., 2003*), which greatly delays mitosis in response to replication stress (*Figure 2—figure supplement 4*), thereby providing more time to complete genome duplication. By carefully examining mitotic progression, we observed that chromosome condensation during prophase was partially defective in the AB cell but not the P1 cell of the *div-1(or148ts)* mutant, and this defect was exacerbated by depletion of TRUL-1 (*Figure 2D*, white arrows; *Figure 2—figure supplement 3C*). These data suggest that the AB cell of the *div-1(or148ts)* mutant enters mitosis before replication has been completed, and that TRUL-1 becomes important for the preservation of chromosome integrity under those conditions.

As an alternative way of inducing a low level of DNA replication stress, we fed worms on plates where bacteria expressing *rnr-1* RNAi were diluted with bacteria containing the empty RNAi vector (*Figure 2—figure supplement 5A*). Whereas the exposure of worms to 100% *rnr-1* RNAi caused a profound defect in condensation during prophase, and severe chromatin bridges during anaphase (*Figure 2—figure supplement 5B*, reproducing the phenotype shown in *Figure 1C*), mitosis progressed normally in most control embryos exposed to 10% *rnr-1* RNAi (*Figure 2—figure supplement 5C*). However, exposure of *trul-1Δ* worms to 10% *rnr-1* RNAi led to impaired condensation during prophase and chromatin bridges during anaphase (*Figure 2—figure supplement 5C–E*), thus reproducing the effect of combining *trul-1* RNAi with the *div-1*(or148ts) mutant (*Figure 2D*). Together with our earlier findings (*Deng et al., 2019*), these data indicated that TRUL-1 helps to preserve genome integrity in *C. elegans*, when cells enter mitosis in the presence of incomplete DNA replication.

## TRAIP is required for FANCD2 focus formation and mitotic DNA repair synthesis at common fragile sites in human cells

To determine whether TRAIP has an evolutionarily conserved role in metazoa, and is required to process sites of incomplete DNA replication during mitosis, we monitored the appearance of mitotic DNA repair synthesis in human cells subjected to a low level of DNA replication stress. We used siRNA to deplete TRAIP in human U2OS cells (*Figure 3A*), and then treated cells with a low dose of the DNA polymerase inhibitor aphidicolin, before synchronisation in the G2-M phase of the cell cycle (*Figure 3B*). Subsequently, cells were released into mitosis in the presence of the nucleoside analogue EdU, and then harvested by mitotic shake off, before detection of FANCD2 foci and sites of EdU incorporation, as described previously (*Minocherhomji et al., 2015*). Strikingly, depletion of TRAIP with either of two different siRNAs suppressed the appearance of both FANCD2 and EdU foci (*Figure 3C–E*). Moreover, this defect was rescued by ectopic expression of an siRNA-resistant version of wild type TRAIP, but not by expression of mutant TRAIP that lacked a functional RING domain, which links the E3 ligase to its cognate E2 enzymes (*Figure 3F–H*). These data indicated that the ubiquitin ligase activity of TRAIP is required upstream of FANCD2 recruitment for mitotic DNA repair synthesis at sites of incomplete DNA replication in human cells. In contrast, depletion of LRR1 in human U2OS cells did not impair MiDAS (*Figure 3—figure supplement 1*).

Consistent with the *C. elegans* data described above, siRNA depletion of TRAIP in human cells led to an increase in anaphase DNA bridges, as well as ultrafine anaphase bridges coated with the PICH ATPase (*Figure 4A–B*). Previous studies have shown that the latter structures generally represent sites where incomplete DNA replication remains unresolved during mitosis (*Bhowmick and Hickson, 2017*). Moreover, cells in the subsequent G1-phase had an increased frequency of 53BP1 bodies (*Figure 4C*), which mark the persistence of sites of incomplete DNA replication from the previous cell cycle (*Lukas et al., 2011*; *Minocherhomji et al., 2015*; *Moreno et al., 2016*). Correspondingly, G1-phase cells also showed increased chromosomal instability in the form of micronuclei (*Figure 4D*). All of these phenotypes were rescued by expression of an siRNA-resistant form of TRAIP (*Figure 4*). These findings confirm that TRAIP is important for metazoan cells to process incomplete DNA replication during mitosis, thus preserving genome integrity.

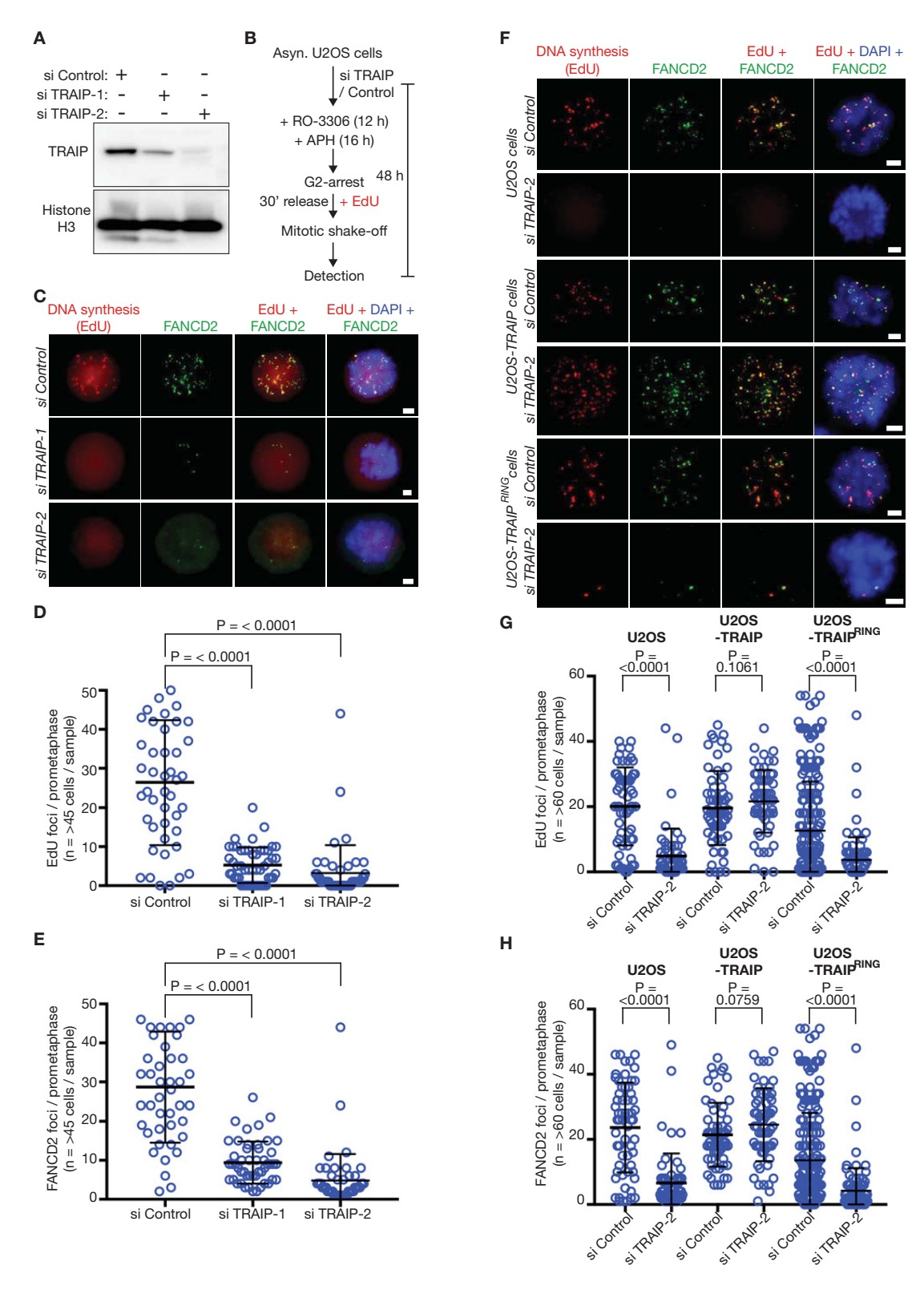

**Figure 3.** The TRAIP E3 ligase is required for mitotic DNA repair synthesis in response to DNA replication stress in human cells. (**A**) Immunoblots illustrating depletion of TRAIP by two specific siRNAs in human U2OS cells. (**B**) Experimental protocol: human U2OS cells were exposed to control siRNA or TRAIP-specific siRNA for 48 hr. During the last 16 hr of siRNA treatment, the cells were treated with the DNA polymerase inhibitor aphidicolin (APH), and the CDK1 inhibitor RO-3306 was then added for the final 12 hr of siRNA treatment to arrest the cells in G2-phase. Subsequently, the cells

*Figure 3 continued on next page*

*Figure 3 continued*

were released from G2-arrest in the absence of RO-3306 for 30 min in the presence of the nucleoside analogue EdU. Mitotic cells were harvested by mitotic shake-off and processed for markers of the mitotic pathway for DNA repair synthesis (foci of EdU incorporation that largely co-localise with foci of the FANCD2 protein). (C) Representative micrographs of 'prometaphase' mitotic cells treated as in (B). The scale bars correspond to 2 µm. (D) Quantification of the number of EdU foci per prometaphase cell. (E) Quantification of the number of FANCD2 foci per prometaphase cell. (F) U2OS control cells, U2OS cells expressing wild type TRAIP, and U2OS cells expressing a RING-mutant of TRAIP were treated as described above. Representative micrographs show EdU foci and FANCD2 foci in prometaphase cells. The scale bars correspond to 2 µm. (G) Quantification of the number of EdU foci per prometaphase cell. (H) Quantification of the number of FANCD2 foci per prometaphase cell. Note that ectopic expression of the TRAIP$^{RING}$ mutant causes a partial dominant negative phenotype (reduced frequency of EdU and FANCD2 foci, even with si Control), as seen previously (*Hoffmann et al., 2016*). For panels D-E and G-H, the broad horizontal lines represent the mean for the entire data set, and the shorter lines above and below denote the standard deviations. P-values were generated via a Mann-Whitney test.

DOI: https://doi.org/10.7554/eLife.48686.011

The following figure supplement is available for figure 3:

**Figure supplement 1.** Depletion of LRR1 does not block MiDAS in human U2-OS cells.

DOI: https://doi.org/10.7554/eLife.48686.012

Whereas S-phase was traditionally thought to end before the G2-phase of the cell cycle, it now appears that DNA replication in some parts of the genome can extend almost to the point of cell division. Such extremely late replicating regions of the genome include common fragile sites, but might also correspond to areas where two converging replication forks have both stalled in a region that lacks intervening dormant origins. Together with recent findings (*Deng et al., 2019*; *Priego Moreno et al., 2019*), our data indicate that TRAIP activity during mitosis promotes replisome disassembly at any remaining replication forks, thus exposing the parental DNA at the junction of the fork (we cannot exclude that TRAIP also has other important targets during mitosis, in addition to CMG). This leads to nucleolytic cleavage and mitotic DNA repair synthesis. In this way, mitotic TRAIP activity brings to an end the period during which conventional DNA replication is possible during the metazoan cell cycle. Although it remains to be demonstrated that TRAIP does indeed drive CMG helicase disassembly during mitosis in human cells, we have recently found that mouse TRAIP is required for mitotic CMG disassembly (Ryo Fujisawa, Fabrizio Villa and Karim Labib, unpublished data), indicating that the role of TRAIP is well conserved in diverse metazoan species.

As noted recently (*Deng et al., 2019*), if cleavage at each of two converging replication forks is specific to the leading strand template, which is normally encircled and thus protected by the CMG helicase, then one intact chromosome would rapidly be restored by gap filling, whereas repair of the other would require the joining of two broken ends. The latter would be associated with a small deletion and with sister chromatid exchange, thus matching the known features of common fragile sites in human cells (*Glover et al., 2017*). Our data indicate that TRAIP preserves viability when metazoan cells enter mitosis with incomplete replication, by promoting a programmed pathway of breakage and repair. This comes at the price of localised genome rearrangements, but avoids the far more damaging random breakage of loci during cell division in the presence of unreplicated DNA.

# Materials and methods

## Key resources table

| Reagent type (species) or resource | Designation | Source or reference | Identifiers | Additional information |
|---|---|---|---|---|
| Strain, strain background (*Caenorhabditis elegans*) | GFP-PSF-1; mCherry-Histone H2B | *Sonneville et al., 2017* PMID: 28368371. | strain name: KAL3 | *psf-1(lab1[gfp::TEV::S-tag::psf-1 + loxP unc-119(+) loxP]); ltIs 37[pie-1p::mCherry::his -58 + unc-119(+)]* |
| Strain, strain background (*Caenorhabditis elegans*) | GFP-CDC-45; mCherry-Histone H2B | *Sonneville et al., 2012* PMID: 22249291. | strain name: TG1754 | *unc-119(ed3) III; gtIs65[pie-1p::gfp::cdc-45 + unc-119(+)]; ltIs37* |

*Continued on next page*

*Continued*

| Reagent type (species) or resource | Designation | Source or reference | Identifiers | Additional information |
|---|---|---|---|---|
| Strain, strain background (*Caenorhabditis elegans*) | GFP-SLD-5; mCherry-Histone H2B | *Sonneville et al., 2012* PMID: 22249291. | strain name: TG1756 | *unc-119(ed3) III; gtls65[pie-1p:: gfp::sld-5 + unc-119(+)]; ltls37* |
| Strain, strain background (*Caenorhabditis elegans*) | GFP-RPA-1; mCherry-Histone H2B | *Sonneville et al., 2012* PMID: 22249291. | strain name: TG2368 | *unc-119(ed3) III; gtls65[pie-1p:: gfp::rpa-1 + unc-119(+)]; ltls37* |
| Strain, strain background (*Caenorhabditis elegans*) | *div-1(or148)* allele | *Encalada et al., 2000* PMID: 11112326. | strain name: EU548 | *div-1(or148[P352L]) III* |
| Strain, strain background (*Caenorhabditis elegans*) | GFP-CAPG-1; mCherry-Histone H2B | *Collette et al., 2011* PMID: 22025633. | strain name: EKM36 | *unc-119(ed3) III; cldls[pie-1p:: gfp::capg-1 + unc-119(+)]; ltls37* |
| Strain, strain background (*Caenorhabditis elegans*) | GFP-KLE-2; mCherry-Histone H2B | *Sonneville et al., 2015* PMID: 26166571. | strain name: TG3828 | *unc-119(ed3) III; gtls3828[pie-1p:: gfp::kle-2 + unc-119(+)]; ltls37* |
| Strain, strain background (*Caenorhabditis elegans*) | *div-1(or148);* GFP-PSF-1; mCherry-Histone H2B | This study | strain name: KAL60 | *div-1(or148) III; psf-1(lab1) II; ltls37* |
| Strain, strain background (*Caenorhabditis elegans*) | *trul-1Δ* | This study | strain name: KAL90 | *trul-1Δ (lab3 [3134 bp deletion])II* |
| Strain, strain background (*Caenorhabditis elegans*) | *trul-1Δ; div-1(or148)* | This study | strain name: KAL139 | *div-1(or148) III; trul-1(lab3) II* |
| Strain, strain background (*Caenorhabditis elegans*) | *trul-1Δ;* GFP-PSF-1; mCherry-Histone H2B | This study | strain name: KAL92 | *trul-1Δ (lab3) psf-1(lab1) II; ltls37* |
| Cell line (*H. sapiens*, Female) | U-2 OS | ATCC | Cat# ATCC HTB-96; RRID:CVCL_0042 | |
| Cell line (*H. sapiens*, Female) | U-2 OS expressing siRNA-resistant human TRAIP (wild type) | *Hoffmann et al., 2016* PMID: 26711499 | | Generated by group of Niels Mailand, by transfection of U-2 OS cells with plasmid expressing siRNA-resistant human TRAIP (wild type) |
| Cell line (*H. sapiens*, Female) | U-2 OS expressing siRNA-resistant human TRAIP (ΔRING - lacking residues 7–50) | *Hoffmann et al., 2016* PMID: 26711499 | | Generated by group of Niels Mailand, by transfection of U-2 OS cells with plasmid expressing siRNA-resistant human TRAIP (ΔRING - lacking residues 7–50) |
| Antibody | Sheep polyclonal, Ce TRUL-1 | This study | C.e. TRUL-1 (70–291) _MRC PPU Reagents and Services: SA607 | Raised against residues 70–291 of C.e. TRUL-1 (MRC PPU Reagents and Services); use at 1:1000 for immunoblotting. |

*Continued on next page*

*Continued*

| Reagent type (species) or resource | Designation | Source or reference | Identifiers | Additional information |
|---|---|---|---|---|
| Antibody | Sheep polyclonal, Ce MCM-2 | *Sonneville et al., 2017* PMID: 28368371. | C.e. MCM-2 (1–222)_MRC PPU Reagents and Services:S750D | Raised against residues 1–222 of C.e. MCM-2 (MRC PPU Reagents and Services); use at 1:3000 for immunoblotting. |
| Antibody | Sheep polyclonal, Ce CDC-45 | *Sonneville et al., 2017* PMID: 28368371. | C.e. CDC-45 (1-222)_MRC PPU Reagents and Services: S782D | Raised against residues 1–222 of C.e. CDC-45 (MRC PPU Reagents and Services); use at 1:500 for immunoblotting. |
| Antibody | Sheep polyclonal, TRAIP | *Hoffmann et al., 2016* PMID: 26711499 | Anti-TRAIP (human) | Raised against full length human TRAIP (Niels Mailand's group); use at 1:250 for immunoblotting. |
| Antibody | Sheep polyclonal, LRR1 | This paper | M.m. LRR1 (1–160)_MRC PPU Reagents and Services:SA279 | Raised against residues 1–160 of M.m. LRR-1 (MRC PPU Reagents and Services); use at 1:250 for immunoblotting. |
| Antibody | Rabbit polyclonal, Anti-GAPDH | Sigma-Aldrich | Cat# PLA0125 | Use at 1:1000 for immunoblotting. |
| Antibody | Rabbit polyclonal, Anti-Histone H3 | Abcam | Cat# ab1791; RRID:AB-302613 | Use at 1:2000 for immunoblotting. |
| Antibody | Rabbit polyclonal, FANCD2 | Novus Biologicals | Cat# NB100-182; RRID:AB-10002867 | Use at 1:400 for immunofluorescence. |
| Antibody | Mouse monoclonal, Anti-PICH, clone 142-26-3 | Millipore | Cat# 04–1540; RRID:AB-10616795 | Use at 1:50 for immunofluorescence. |
| Antibody | Mouse monoclonal, 53BP1 (6B3E10) | Santa Cruz Biotechnology | Cat# sc-517281 | Use at 1:500 for immunofluorescence. |
| Antibody | Goat polyclonal, Anti-Mouse IgG (whole molecule)–peroxidase conjugated | Sigma Aldrich | Cat# A4416; RRID:AB_258167 | Use at 1:2000 for immunoblotting. |
| Antibody | Goat polyclonal Anti-Rabbit IgG (whole molecule), F(ab')two fragment—Peroxidase conjugated | Sigma Aldrich | Cat# A6667; RRID: AB_258307 | Use at 1:2000 for immunoblotting. |
| Antibody | Donkey polyclonal, Anti-Sheep IgG (whole molecule)–Peroxidase conjugated | Sigma Aldrich | Cat# A3415; RRID:AB_258076 | Use at 1:2000 for immunoblotting. |
| Antibody | Goat polyclonal, anti-Rabbit IgG (H+L) Cross-Adsorbed Secondary Antibody, Alexa Fluor 488 | Invitrogen | Cat# A-11008; RRID: AB_143165 | Use at 1:1000 for immunofluorescence. |

*Continued on next page*

*Continued*

| Reagent type (species) or resource | Designation | Source or reference | Identifiers | Additional information |
|---|---|---|---|---|
| Antibody | Goat polyclonal, anti-Rabbit IgG (H+L) Cross-Adsorbed Secondary Antibody, Alexa Fluor 568 | Invitrogen | Cat# A-11011; RRID: AB_143157 | Use at 1:1000 for immunofluorescence. |
| Antibody | Goat polyclonal, anti-Mouse IgG (H+L) Highly Cross-Adsorbed Secondary Antibody, Alexa Fluor 488 | Invitrogen | Cat# A-11029; RRID:AB_138404 | Use at 1:1000 for immunofluorescence. |
| Antibody | Donkey polyclonal, anti-Mouse IgG (H+L) Highly Cross-Adsorbed Secondary Antibody, Alexa Fluor 568 | Invitrogen | Cat# A-10037; RRID:AB_2534013 | Use at 1:1000 for immunofluorescence. |
| Sequence-based reagent | oligo to genotype *trul-1Δ*: Primer a | This study | | 5'-ACACCATAGCGATTGTTTCCGG-3' |
| Sequence-based reagent | oligo to genotype *trul-1Δ*: Primer b | This study | | 5'-CCGGTGGTTTTTCAGCTTCTCC-3' |
| Sequence-based reagent | oligo to genotype *trul-1Δ*: Primer c | This study | | 5'-GATTCGTGTGGATTTCTGCGGT-3' |
| Sequence-based reagent | ON-TARGETplus *LRR1* siRNA | Dharmacon | Cat# LQ-016820–01 | |
| Sequence-based reagent | *TRAIP* siRNA # 1 (5'-GAACCAUUAU CAAUAAGCU-3') | Sigma-Aldrich | Custom synthesised | *Hoffmann et al., 2016* PMID: 26711499 |
| Commercial assay or kit | Click-iT EdU Alexa Fluor 594 Imaging Kit | ThermoFisher Scientific | Cat# C10339 | |
| Chemical compound, drug | Aphidicolin | Sigma-Aldrich | Cat# A0781 | |
| Chemical compound, drug | RO-3306 | Millipore | Cat# 217699 | |
| Software, algorithm | Image processing using ImageJ software | ImageJ software (National Institutes of Health) | | |
| Software, algorithm | Prism data analysis software | Graphpad | | |
| Software, algorithm | Image processing using Fiji software | ImageJ | | |
| Other | Lipofectamine RNAimax | Life Technologies | Cat#13778075 | |

## *C. elegans* maintenance

The *C. elegans* strains were maintained according to standard procedures (*Brenner, 1974*) and were grown on 'Nematode Growth Medium' (NGM: 3 g / L NaCl; 2.5 g / L peptone; 20 g / L agar; 5 mg / L cholesterol; 1 mM $CaCl_2$; 1 mM $MgSO_4$; 2.7 g / L $KH_2PO_4$; 0.89 g / L $K_2HPO_4$). The following worm strains were used: KAL3 *psf-1(lab1[gfp::TEV::S-tag::psf-1 + loxP unc-119(+) loxP]); ltIs37[pie-1p:: mCherry::his-58 + unc-119(+)]* (*Sonneville et al., 2017*), TG1754 *unc-119(ed3) III; gtIs65[pie-1p::gfp:: cdc-45 + unc-119(+)]; ltIs37* (*Sonneville et al., 2012*), TG1756 *unc-119(ed3) III; gtIs65[pie-1p::gfp:: sld-5 + unc-119(+)]; ltIs37* (*Sonneville et al., 2012*), TG2368 *unc-119(ed3) III; gtIs65[pie-1p::gfp::rpa-*

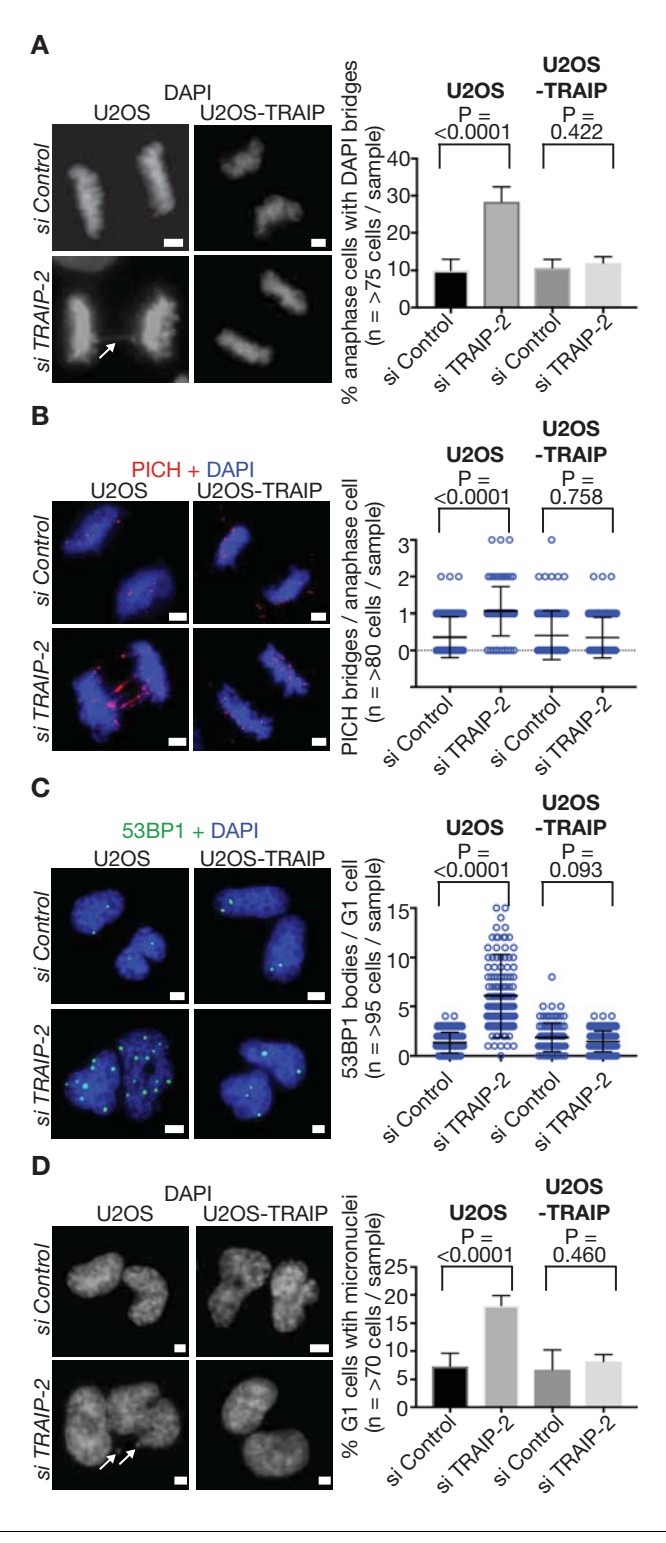

**Figure 4.** TRAIP is required to process sites of incomplete DNA replication and preserve genome integrity in response to DNA replication stress in human cells. (**A**) U2OS control cells and those expressing wild type TRAIP were treated as described in *Figure 3B*, except that cells were re-seeded onto plates for 15 min after mitotic shake-off, before detection of DAPI-positive DNA bridges in anaphase cells (white arrow denotes bridge - the data are quantified in the right-hand panel). (**B**) As per (**A**), except that ultra-fine DNA bridges were monitored by immunofluorescence detection of the PICH ATPase. Chromosomal DNA was detected by DAPI-staining. Note that *Figure 4 continued on next page*

*Figure 4 continued*

the ultra-fine bridges are DAPI-negative. A quantification of the data is presented in the right-hand panel. (**C**) An analogous experiment to those above was performed, except that cells were re-seeded for 2.5 hr before detection of the 53BP1 protein by immunofluorescence in newly-born G1 cells. The number of 53BP1 bodies per G1-phase cell is quantified in the right-hand panel. (**D**) As per (**C**), except the number of micronuclei per G1-phase cell (indicated by white arrows) was determined by DAPI-staining. Statistics presented in this figure were calculated as described above for *Figure 3*. The scale bars correspond to 2 μm.

DOI: https://doi.org/10.7554/eLife.48686.013

1 + *unc-119(+)*]; *ltIs37* (***Sonneville et al., 2012***), EU548 *div-1(or148[P352L]) III* (***Encalada et al., 2000***), EKM36 *unc-119(ed3) III; cldIs[pie-1p::gfp::capg-1 + unc-119(+)]; ltIs37* (***Collette et al., 2011***), TG3828 *unc-119(ed3) III; gtIs3828[pie-1p::gfp::kle-2 + unc-119(+)]; ltIs37* (***Sonneville et al., 2015***).

The following worm strains were made for this study: KAL60 *div-1(or148) III; psf-1(lab1) II; ltIs37* was generated by crossing EU548 with KAL3. KAL90 *trul-1Δ (lab3 [3134 bp deletion])II* was generated using the CRISPR-CAS9 system (NemaMetrix) and *trul-1Δ* worms were out-crossed eight times with the N2 strain. KAL139 *div-1(or148) III; trul-1(lab3) II* was generated by crossing KAL90 with EU548. KAL92 *trul-1Δ (lab3) psf-1(lab1) II; ltIs37* resulted from crossing KAL90 and KAL3.

## *C. elegans* progeny viability assay

L4 worms (from a stock maintained at 15℃) were fed at 15℃ for 72 hr, or at 20℃ for 48 hr, and five adult worms were then transferred to a test plate for 150 min in order to lay embryos. The adult worms were then removed and the total number of embryos was counted (typically around 80 embryos). Subsequently, the number of embryos that developed into viable adults was determined and the embryonic viability was expressed as the ratio between the number of viable embryos and the total number of embryos.

## RNA interference in *C. elegans*

RNAi was performed by feeding worms with RNAse III-deficient HT115 bacteria transformed with an L4440-derived plasmid that express double-stranded RNA (dsRNA) (***Timmons and Fire, 1998***). Bacterial culture (grown to OD600 = 1) was supplemented with 1 mM IPTG to express dsRNA. To titrate *rnr-1* RNAi, we mixed bacteria expressing *rnr-1* dsRNA with bacteria containing the empty vector. For example to obtain 10% *rnr-1* RNAi, 1 vol of culture of bacteria expressing *rnr-1* dsRNA was mixed with 9 volumes of control bacterial culture. Subsequently, 400 μl of bacteria were spread on a 6 cm RNAi plate (3 g / L NaCl, 20 g / L agarose, 5 mg / L cholesterol, 1 mM $CaCl_2$, 1 mM $MgSO_4$, 2.7 g / L $KH_2PO_4$, 0.89 g / L $K_2HPO_4$, 1 mM IPTG and 100 mg / L Ampicillin) and RNAi was induced overnight at room temperature.

The plasmids expressing dsRNA were made by cloning PCR products amplified from cDNA into the vector L4440 as described previously: *rnr-1* and *rpa-1* RNAi in ***Sonneville et al. (2015)***; *npl-4* and *lrr-1* RNAi in ***Sonneville et al. (2017)***; *trul-1* RNAi in ***Deng et al. (2019)***. To target simultaneously *rnr-1 + npl-4*, *rpa-1 + npl-4*, *rnr-1 + lrr-1* and *rpa-1 + lrr-1*, DNA fragments corresponding to each gene were cloned contiguously into a single L4440 plasmid. The empty L4440 plasmid was used as a negative control throughout the RNAi experiments in this study.

## Microscopy

Worms at the larval L4 stage were incubated on 6 cm RNAi feeding plates for 48 hr at 20℃. In the experiment using the *div1(or148ts)* mutant (*Figure 2D–E*), all worms were shifted to the restrictive temperature of 25℃ for 24 hr, prior to imaging. For each experimental condition, five embryos were dissected from different worms in M9 medium (6 g / L $Na_2HPO_4$, 3 g / L $KH_2PO_4$, 5 g / L NaCl, 0.25 g / L $MgSO_4$) and mounted together on a 2% agarose pad. Embryos were then recorded simultaneously at 24℃, taking images every 10 s with 75% laser power and 200 milliseconds exposure time. Time lapse images were acquired using an Zeiss Cell Observer SD microscope with a Yokogawa CSU-X1 spinning disk, HAMAMATSU C13440 camera, fitted with a PECON incubator, using the ZEN blue software, and then processed with ImageJ software (National Institutes of Health) as previously described (***Sonneville et al., 2017***).

## Quantification of signal intensity from *C. elegans* embryos

The GFP and mCherry signal intensity was evaluated during metaphase using ImageJ software (NIH). For each metaphase set of chromosomes, an area of fixed size was selected around the chromatin, and the integrated density was determined (the sum of the intensity of all pixels in the area). Subsequently, the data were adjusted for the background signal, by subtracting the integrated density of an area of equivalent size from the cytoplasm of the same embryo. The average signal intensity and standard deviation were then determined for five embryos per condition.

## Genotyping worms

10 worms were lysed in 10 µl of buffer (1X PCR buffer, 1 mM MgCl2, 0.45% Tween-20) supplemented with 1.7 µg of Proteinase K at 60˚C for 1 hr. The protease was inactivated at 95˚C for 15 min and PCR was then performed using 0.5 µl lysed worms in a 25 µl reaction volume using Phusion polymerase (Biolabs). The following oligonucleotides were used to genotype the *trul-1Δ* and wild-type worms (as shown in *Figure 2—figure supplement 1*): Primer a 5'-ACACCATAGCGATTGTTTCCGG-3', Primer b 5'-CCGGTGGTTTTTCAGCTTCTCC-3' and Primer c 5'-GATTCGTGTGGATTTCTGCGGT-3'.

## Extracts of worm embryos

Gravid adult worms from two 9 cm NGM plates were collected in M9 buffer and bleached with 10 ml of bleaching solution (for 100 ml: 36.5 ml $H_2O$, 45.5 ml 2 N NaOH and 18 ml ClNaO 4%) for about 8 min to release the embryos. The embryos were then washed twice with M9 buffer, suspended in 150 µl of M9 buffer and frozen on dry ice. Pellets were thawed and supplemented with 150 µl of 40% Trichloroacetic acid and 300 µl of glass beads. The cells were then broken by vortexing for one minute and the supernatant was collected. The beads were washed with 300 µl of 5% Trichloroacetic acid and the supernatant was added to the previous one. The extracts were then spun at 3000 rpm for 10 min and the pellet was re-suspended in 'high pH Laemmli buffer' (Laemmli buffer supplemented with 0.15 M Tris base).

## Antibodies

The *C. elegans* TRUL-1 protein was detected with a sheep polyclonal antibody that was raised against amino acids 70–291 (MRC PPU reagents; SA607), whilst the anti-MCM-2 and anti-CDC-45 antibodies were described previously (*Sonneville et al., 2017*).

Primary antibodies specific for human PICH (1:50, 04–1540; Millipore), 53BP1 (1:500, sc-517281; Santa Cruz Biotechnology), FANCD2 (1:500, Novus Biologicals; NB100-182), TRAIP (1:250; Hoffmann et al – see Key Resources Table), GAPDH (1:1000, Sigma Aldrich; PLA0125), Histone H3 (1:2000, abcam; ab1791), were obtained from the indicated supplier. The corresponding secondary antibodies were: anti-mouse IgG (whole molecule peroxidase conjugate, 1:2,000, A4416; Sigma Aldrich), anti-rabbit IgG (whole molecule, F(ab')two fragment–peroxidase antibody produced in goat (1:2,000, A6667; Sigma Aldrich). Secondary antibodies used for immunofluorescence analysis of human cells were the Alexa Fluor 488 goat anti-rabbit IgG (1:1000, A-11008; Invitrogen), Alexa Fluor 568 goat anti-rabbit IgG (1:1000, A- A-11011; Invitrogen), Alexa Fluor 488 goat anti-mouse IgG (1:1000, A-11029; Invitrogen) and Alexa Fluor 568 donkey anti-mouse IgG (1:1000, A-10037; Invitrogen).

## Human cell culture

U2OS osteosarcoma cells were maintained in Dulbecco's modified Eagle's medium (DMEM; Invitrogen, 11960), supplemented with 10% fetal bovine serum (FBS; Invitrogen, 10500) and 1% penicillin-streptomycin-glutamine (P-S-G; Invitrogen, 10378016). U2OS cells stably expressing either wildtype TRAIP (U2OS_TRAIP) or RING domain deleted TRAIP (U2OS_ΔRING TRAIP) were maintained similarly, except that the medium was supplemented with 5 µg / ml Blasticidin (Thermo Fisher Scientific, R21001) and 0.2 mg / ml Hygromycin B (Thermo Fisher Scientific, 10687010). All cells were maintained at 37˚C in a humidified atmosphere containing 5% CO2. U2OS cells were obtained from the ATCC, and all cell lines were routinely tested for the absence of mycoplasma contamination (using MycoAlert; Lonza Group Ltd, LT07-318).

## Human cell synchronisation

Asynchronously growing cells were treated with 7 μM RO-3306 (CDK1 inhibitor) for 12 hr to synchronise/arrest at the G2 phase of the cell cycle. The arrested cells were washed 3 times with warm DMEM medium and were then allowed to progress synchronously for 30 mins into prometaphase. The prometaphase cells were shaken off to be reseeded onto poly-lysine-coated slides, and were then either fixed or allowed to progress for 20 mins (anaphase) or 2.5 hr (G1 phase).

## RNA interference treatment of human cells

Human TRAIP was depleted with either of two previously validated siRNAs (5'-GAACCAUUAUCAA UAAGCU-3'; 5'-CCGUGAUGAUAUUGAUCUCAA-3'). All siRNAs were custom synthesised by Sigma Aldrich with a 3'dTdT addition. In all transfection experiments, the 'Silencer-Select Control siRNA 2' (4390846, Ambion) was used as a negative control. For transient transfections, cells were seeded 12 hr prior to transfection in culture medium supplemented with FBS. The transfection was performed using lipofectamine RNAi Max reagent (Thermo Fisher Scientific, 13778030) according to the manufacturer's protocol.

## EdU labelling and detection in human cells

EdU labelling and detection was performed as described previously (*Minocherhomji et al., 2015*). Click-IT chemistry was performed according to the manufacturer's instructions, but with a final concentration of 10X of the Click-IT EdU buffer additive (Life Technologies).

## Immunofluorescence

Cells on glass cover slips or poly-lysine glass slides were fixed and permeabilised using cold PTEMF buffer (20 mM PIPES, pH 6.8, 10 mM EGTA, 0.2% Triton X-100, 1 mM $MgCl_2$ and 4% formaldehyde) for 20 mins at room temperature followed by blocking with PBSAT (0.5% Triton X in 3% BSA dissolved in PBS) for 20 mins at room temperature. Samples were then incubated with primary antibodies overnight at 4°C. Excess unbound primary antibody was removed by washing four times with PBSAT for 15 mins each. Samples were then incubated with secondary antibodies for 60 mins at room temperature, followed by four washes with PBSAT for 15 mins each. Air-dried slides or coverslips were mounted using VectaShield mounting medium with DAPI (Vector Laboratories). Images were captured using an Olympus BX63 microscope.

## Statistics and reproducibility

The experiments were not randomised, no statistical method was used to predetermine sample size, and the investigators were not blinded to allocation during experiments and outcome assessment.

For *C. elegans* microscopy experiments, at least five embryos were analysed and seen to behave similarly, unless indicated otherwise. For *C. elegans* progeny viability assays, three biological replicates (independent experiments) were performed and the data represent the mean embryonic viability and the standard deviation for each condition. The PCR experiment in *Figure 2—figure supplement 1B* was performed 10 times, whereas the immunoblotting experiments in *Figure 2—figure supplement 1B* were performed 3 times, and those in *Figure 3A* were performed twice.

For work with human cells, at least 45–95 cells were examined per sample in the microscopy experiments (as indicated in *Figures 3–4*), and graphical and statistical analysis was carried out using 'Prism' software (GraphPad). At least three biological replicates were performed for each data set, which were then used to generate mean values and standard deviations. Statistical significance of these data was calculated using a Mann–Whitney U test (to minimise the effect of outliers in the data).

## Acknowledgements

We thank the Medical Research Council for funding (core grant MC_UU_12016/13 to KL), together with Cancer Research UK (Programme Grant C578/A24558), the Wellcome Trust (reference 102943/Z/13/Z for an Investigator award to KL), the Danish National Research Foundation (DNRF115; to IDH), the Danish Medical Research Council (DFF-4004-00155B; a postdoctoral fellowship grant to RB), and the European Research Council (grant no. 616236 to NM). We thank Johannes Walter and

David Pellman for sharing unpublished data, Ryo Fujisawa and Julia Greiwe for assistance in the early stages of the *C. elegans* TRUL-1 experiments, and MRC PPU Reagents and Services (https://mrcppureagents.dundee.ac.uk) for the anti-TRUL-1 antibody.

## Additional information

### Funding

| Funder | Grant reference number | Author |
|---|---|---|
| Medical Research Council | MC_UU_12016/13 | Remi Sonneville<br>Karim Labib |
| Cancer Research UK | C578/A24558 | Remi Sonneville<br>Karim Labib |
| Wellcome | 102943/Z/13/Z | Karim Labib |
| Danish National Research Foundation | DNRF115 | Ian D Hickson |
| Danish Medical Research Council | DFF-4004-00155B | Rahul Bhowmick |
| H2020 European Research Council | 616236 | Niels Mailand |

The funders had no role in study design, data collection and interpretation, or the decision to submit the work for publication.

### Author contributions

Remi Sonneville, Conceptualization, Formal analysis, Validation, Investigation, Visualization, Methodology, Writing—original draft; Rahul Bhowmick, Formal analysis, Validation, Investigation, Visualization, Methodology, Writing—review and editing; Saskia Hoffmann, Resources, Formal analysis, Investigation, Methodology, Writing—review and editing; Niels Mailand, Resources, Formal analysis, Supervision, Funding acquisition, Investigation, Methodology, Writing—review and editing; Ian D Hickson, Resources, Formal analysis, Supervision, Funding acquisition, Validation, Investigation, Methodology, Writing—review and editing; Karim Labib, Conceptualization, Resources, Formal analysis, Supervision, Funding acquisition, Validation, Writing—original draft, Project administration

### Author ORCIDs

Karim Labib https://orcid.org/0000-0001-8861-379X

### Decision letter and Author response

Decision letter https://doi.org/10.7554/eLife.48686.016
Author response https://doi.org/10.7554/eLife.48686.017

## Additional files

### Supplementary files
• Transparent reporting form
DOI: https://doi.org/10.7554/eLife.48686.014

### Data availability
All data generated or analysed during this study are included in the manuscript and supporting files.

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
