## [Decision Letter]

Thank you for submitting your article "TRAIP drives replisome disassembly and mitotic DNA repair synthesis at sites of incomplete DNA replication" for consideration by *eLife*. Your article has been reviewed by three peer reviewers, one of whom is a member of our Board of Reviewing Editors, and the evaluation has been overseen by Jessica Tyler as the Senior Editor. The following individual involved in review of your submission has agreed to reveal their identity: Dirk Remus (Reviewer #3).

The reviewers have discussed the reviews with one another and the Reviewing Editor has drafted this decision to help you prepare a revised submission.

Summary:

This study investigates the role of TRAIP, a ubiquitin ligase important for genome maintenance, in the cellular response to replication stress. The data show that TRAIP is required for the removal of replisomes from mitotic chromosomes in early *C. elegans* embryos, in which normal DNA replication fork progression was impaired by RPA or RNR depletion, and that this defect correlates with reduced viability and chromosome segregation defects. The data in human cells show that TRAIP is required for mitotic DNA repair synthesis and for the prevention of anaphase chromosome bridges, which by extension suggests a requirement of stalled replisome disassembly for the completion of DNA replication in mitosis after replication stress, consistent with the models derived from the *Xenopus* system earlier. This study is of significant importance for the DNA replication and repair fields, providing critical experimental tests of existing hypotheses. A few suggestions are included below to make the manuscript stronger.

Essential revisions:

- It is necessary to quantify the intensity of the protein signal to appreciate differences between mutants that show the same phenotype but to different extents in Figure 1. That would be the case of CMG persistence on mitotic cells in *npl-4* RNAi and *npl-4* + *rnr-1* RNAi cells or the barely detectable mCherry-H2B in *rrn-1* RNAi and *npl-4* + *rnr-1* RNAi cells. Authors could also consider quantifying the amount of CMG components on chromatin during S phase should be quantified to exclude that possibility.

- The authors show convincingly in Figure 2A and B that TRUL-1, but not LRR-1, is necessary to disassemble CMG from chromatin in the presence of replicative stress in *rpa-1* or *rnr-1* RNAi cells. However, this comparison is not made in Figure 2D in *div-1* mutant cells. This comparison is important to confirm previous observations.

- In Figure 3, authors clearly show the role of TRAIP in MiDAS. However, it would be important to link this phenomenon to the CMG disassembly from the chromatin to prove that upon these experimental conditions in U2OS cells, CMG components are retained in chromatin. In addition, TRAIP, but not LRR1, is involved in CMG disassembly when replication is not completed. In that sense, it is also important to show that LRR1 absence does not reduce MiDAS to confirm the observations made in *C. elegans*.

- There is no evidence that human TRAIP leads to helicase stabilisation on unreplicated or even fully replicated chromatin. Therefore, it is not clear whether the phenotypes observed after TRAIP siRNA in human cells are due to helicase issues or another function of this protein, such as fork progression (Harley et al., 2016 and Hoffman et al., 2016). This could limit the impact of the study, so please discuss how this uncertainty affects your conclusions.

- In *C. elegans*, TRAIP is required (with *lrr-1*) for mitotic helicase unloading. The authors show here that failure to unload the helicase in normal cell cycles leads to anaphase bridges (Figure 2—figure supplement 2C). It is therefore unclear whether the genome instability that arises from loss of TRAIP in *C. elegans* or humans is due to incomplete replication, failure to unload the helicase or some other function of this ligase. It would be good to clarify this further or modify the message accordingly.

- We appreciate that it is difficult to get a graded amount of replication stress in *C. elegans*, but the three types of stress here (RPA1 Rnai, Rnr1 RNAi and *div-1*) all give different results with helicase unloading and Trul-1 loss (Figure 2A, B and D). A useful experiment would be to use a titration of a replication inhibitor such as HU (see Brauchle et al., 2003).

---

## [Author Response]

Essential revisions:- It is necessary to quantify the intensity of the protein signal to appreciate differences between mutants that show the same phenotype but to different extents in Figure 1. That would be the case of CMG persistence on mitotic cells in npl-4 RNAi and npl-4 + rnr-1 RNAi cells or the barely detectable mCherry-H2B in rrn-1 RNAi and npl-4 + rnr-1 RNAi cells. Authors could also consider quantifying the amount of CMG components on chromatin during S phase should be quantified to exclude that possibility.

We now present quantification of the GFP-PSF-1, GFP-SLD-5 and GFP-CDC-45 signals on metaphase chromatin (Figure 1—figure supplement 1D, G-H), expressed as a percentage of the *npl-4* RNAi control (which we set to 100%).

Moreover, Figure 1—figure supplement 1E now presents quantification of the mCherry-Histone-H2B signal during metaphase, expressed as a percentage of the no RNAi control (which we set to 100%). This illustrates the point that we made in the text, namely that the mCherry-Histone-H2B signal is specifically reduced to a barely detectable level upon co-depletion of *rnr-1* and *npl-4*.

It was not possible to quantify the amount of CMG on chromatin during S-phase in these experiments, as the assay is dependent upon chromatin condensation during mitosis (otherwise we would just be quantifying the total nuclear signal). Though quantification during S-phase would be very interesting, this would require future development of alternative approaches, such as FRET between two different CMG components that only come together at replication forks.

- The authors show convincingly in Figure 2A and B that TRUL-1, but not LRR-1, is necessary to disassemble CMG from chromatin in the presence of replicative stress in rpa-1 or rnr-1 RNAi cells. However, this comparison is not made in Figure 2D in div-1 mutant cells. This comparison is important to confirm previous observations.

As noted by the reviewers, the assays in Figure 2A-B show that TRUL-1, but not LRR-1, is required for CMG disassembly upon replication stress (RNAi to *rpa-1* or *rnr-1*).

The experiments in Figure 2C-E then explore the physiological significance of the TRUL-1 pathway, by combining *trul-1∆* with a mild form of replication stress produced by the *div-1(or148)* mutation, and monitoring synthetic phenotypes (synthetic lethality in Figure 2C, synthetic condensation / replication defect in Figure 2D, and synthetic induction of chromatin bridges in Figure 2E).

The key point for these experiments is that deletion of *trul-1* is not lethal (our data in this manuscript), and does not itself induce DNA replication or perturb cell cycle progression (Figure 2—figure supplement 4). In contrast, inactivation of *lrr-1* causes embryonic lethality and DNA replication stress (Merlet et al., 2010). This makes it much harder to interpret the impact of *lrr-1* RNAi on viability and genome integrity in combination with *div-1(or148)*.

In our unpublished work, we have shown that diluted *lrr-1* RNAi does not cause synthetic lethality with *div-1(or148*), in contrast to the combination of *trul-1∆* with *div-1(or148)* (we diluted *lrr-1* RNAi to a point that we previously showed is viable in wild type, but is lethal in combination with depletion of a factor called UBXN-3: Sonneville et al., 2017). Conversely, 100% *lrr-1* RNA causes a synthetic DNA replication stress phenotype with *div-1(or148)*. However, both of these results are very hard to interpret, for the reasons given above.

So we prefer in the revised manuscript to discuss briefly these complications, citing the past work of Merlet et al., 2010, from the group of Lionel Pintard. We now do so on in the subsection “TRAIP preserves genome integrity and promotes survival in response to DNA replication stress”.

- In Figure 3, authors clearly show the role of TRAIP in MiDAS. However, it would be important to link this phenomenon to the CMG disassembly from the chromatin to prove that upon these experimental conditions in U2OS cells, CMG components are - in chromatin. In addition, TRAIP, but not LRR1, is involved in CMG disassembly when replication is not completed. In that sense, it is also important to show that LRR1 absence does not reduce MiDAS to confirm the observations made in C. elegans.

We now show in Figure 3—figure supplement 1 that depletion of LRR1 in human U2OS cells does not reduce MiDAS (subsection “TRAIP is required for FANCD2 focus formation and mitotic DNA repair synthesis at common fragile sites in human cells”).

Though we agree that it will be important to develop assays with which to monitor CMG disassembly in human cells, this was not possible within the two-month review period for this manuscript. Nevertheless, in another unpublished project with mouse ES cells, we have found in the last few months that LRR1 mediates the S-phase pathway of CMG helicase disassembly, whereas TRAIP drives mitotic CMG disassembly. This indicates that the roles of LRR1 and TRAIP are well conserved amongst diverse metazoa including mammals. So it is highly likely that the role of human TRAIP in MiDAS does indeed reflect its role in a mitotic pathway of CMG helicase disassembly.

We now refer to these data in the subsection "TRAIP is required for FANCD2 focus formation and mitotic DNA repair synthesis at common fragile sites in human cells

- There is no evidence that human TRAIP leads to helicase stabilisation on unreplicated or even fully replicated chromatin. Therefore, it is not clear whether the phenotypes observed after TRAIP siRNA in human cells are due to helicase issues or another function of this protein, such as fork progression (Harley et al., 2016 and Hoffman et al., 2016). This could limit the impact of the study, so please discuss how this uncertainty affects your conclusions.

As noted above, we now refer to unpublished data in mouse ES cells, showing that mammalian TRAIP is responsible for a mitotic pathway that extracts the CMG helicase from chromatin (subsection "TRAIP is required for FANCD2 focus formation and mitotic DNA repair synthesis at common fragile sites in human cells").

- In C. elegans, TRAIP is required (with lrr-1) for mitotic helicase unloading. The authors show here that failure to unload the helicase in normal cell cycles leads to anaphase bridges (Figure 2—figure supplement 2C). It is therefore unclear whether the genome instability that arises from loss of TRAIP in C. elegans or humans is due to incomplete replication, failure to unload the helicase or some other function of this ligase. It would be good to clarify this further or modify the message accordingly.

In both worms and human cells lacking TRAIP, we only observe genome instability in combination with DNA replication stress. In human cells lacking TRAIP and subjected to replication stress, the accumulation during mitosis of DNA bridges coated with the PICH ATPase, together with the accumulation of 53BP bodies in the subsequent G1-phase, indicate that genome instability results from a failure to complete DNA replication (likely due to the failure of MiDAS, in turn due to the failure to disassemble CMG during mitosis). At this stage, we cannot exclude that TRAIP also has other functions when cells enter mitosis with incomplete DNA replication, and we now discuss this fact in the subsection “TRAIP is required for FANCD2 focus formation and mitotic DNA repair synthesis at common fragile sites in human cells”.

- We appreciate that it is difficult to get a graded amount of replication stress in C. elegans, but the three types of stress here (RPA1 Rnai, Rnr1 RNAi and div-1) all give different results with helicase unloading and Trul-1 loss (Figure 2A, B and D). A useful experiment would be to use a titration of a replication inhibitor such as HU (see Brauchle et al., 2003).

We agreed that this was an interesting idea, but Brauchle et al. injected worms with HU, and we felt that this would have made it difficult to achieve reproducible data in a titration experiment. Therefore, we took an alternative approach to generating a milder form of DNA replication stress, by titration of *rnr-1* RNAi. As shown in Figure 2—figure supplement 5, 10% *rnr-1* RNAi in *trul-1∆* worms produced a similar phenotype to the combination of *trul-1* RNAi with the *div-1(or148ts)* mutant, leading to an accumulation of chromatin bridges and a condensation defect in the AB cell. These data are now discussed on in the subsection “TRAIP preserves genome integrity and promotes survival in response to DNA replication stress”.